# Caffeine Intake, Plasma Caffeine Level, and Kidney Function: A Mendelian Randomization Study

**DOI:** 10.3390/nu15204422

**Published:** 2023-10-18

**Authors:** Alice Giontella, Roxane de La Harpe, Héléne T. Cronje, Loukas Zagkos, Benjamin Woolf, Susanna C. Larsson, Dipender Gill

**Affiliations:** 1Department of Clinical Sciences Malmö, Lund University, Jan Waldenströms Gata 35 Malmö, 214 28 Malmo, Sweden; alice.giontella@med.lu.se; 2Unit of Internal Medicine, Department of Medicine, University Hospital of Lausanne, Rue du Bugnon 21, 1011 Lausanne, Switzerland; roxane.de-la-harpe@chuv.ch; 3Department of Public Health, Section of Epidemiology, University of Copenhagen, 1165 Copenhagen, Denmark; toinet.cronje@sund.ku.dk; 4Department of Epidemiology and Biostatistics, School of Public Health, Imperial College London, London SW7 2BX, UK; l.zagkos@imperial.ac.uk; 5Medical Research Council Biostatistics Unit, University of Cambridge, Cambridge CB2 0SR, UK; benjamin.woolf@bristol.ac.uk; 6Medical Research Council Integrative Epidemiology Unit, University of Bristol, Bristol BS8 2BN, UK; 7Department of Psychological Science, University of Bristol, Bristol BS8 1TU, UK; 8Unit of Medical Epidemiology, Department of Surgical Sciences, Uppsala University, 751 85 Uppsala, Sweden; susanna.larsson@ki.se; 9Unit of Cardiovascular and Nutritional Epidemiology, Institute of Environmental Medicine, Karolinska Institutet, 171 77 Stockholm, Sweden

**Keywords:** caffeine level, caffeine intake, genetically predicted coffee consumption, causal inference, Mendelian randomization, kidney function, estimated glomerular filtration rate

## Abstract

Caffeine is a psychoactive substance widely consumed worldwide, mainly via sources such as coffee and tea. The effects of caffeine on kidney function remain unclear. We leveraged the genetic variants in the *CYP1A2* and *AHR* genes via the two-sample Mendelian randomization (MR) framework to estimate the association of genetically predicted plasma caffeine and caffeine intake on kidney traits. Genetic association summary statistics on plasma caffeine levels and caffeine intake were taken from genome-wide association study (GWAS) meta-analyses of 9876 and of >47,000 European ancestry individuals, respectively. Genetically predicted plasma caffeine levels were associated with a decrease in estimated glomerular filtration rate (eGFR) measured using either creatinine or cystatin C. In contrast, genetically predicted caffeine intake was associated with an increase in eGFR and a low risk of chronic kidney disease. The discrepancy is likely attributable to faster metabolizers of caffeine consuming more caffeine-containing beverages to achieve the same pharmacological effect. Further research is needed to distinguish whether the observed effects on kidney function are driven by the harmful effects of higher plasma caffeine levels or the protective effects of greater intake of caffeine-containing beverages, particularly given the widespread use of drinks containing caffeine and the increasing burden of kidney disease.

## 1. Introduction

Caffeine, a central nervous system stimulant, is present in coffee beans, tea leaves, and cacao beans and is commonly added to energy and carbonated drinks as well as analgesic drugs [1]. By inhibiting phosphodiesterase enzyme, antagonizing adenosine receptors, and activating ryanodine receptors, the adverse effects of caffeine on kidney function and structure have been postulated [2,3].

Increasing evidence has indicated that caffeine may have detrimental effects on kidney function, although the literature remains controversial. Animals’ investigations have shown an increase in proteinuria and renal vascular resistance, leading to marked renal failure [3,4]. In patients suffering from autosomal dominant polycystic kidney disease, studies have reported an increased risk of cyst enlargement with increasing caffeine intake [5,6], while others did not show such consequences [7,8]. Observational studies and meta-analyses observed a lower risk of developing chronic kidney disease (CKD) with increasing coffee consumption [9,10,11,12,13,14]. However, these studies suffered potential bias related to their observational design and assessed caffeine exposure as a self-reported intake of cups of coffee/day, not accounting for the wide interindividual variation in the metabolism of caffeine that could modify any associations. Indeed, a recent study in 1180 adults showed that the risks of albuminuria and hyperfiltration were over 2-fold higher among slow metabolizers of caffeine who consumed more than three cups of coffee per day compared to those with low coffee intake, while there was no difference among fast metabolizers [15]. Over 95% of caffeine in humans is metabolized via cytochrome P450 1A2 (CYP1A2), and the gene expression of this enzyme is regulated via the aryl hydrocarbon receptor (AHR) [16,17]. The heritability of coffee consumption was estimated at 36% to 58% [18].

Here, we leveraged the genetic variants in the *CYP1A2* and *AHR* genes via the two-sample Mendelian randomization (MR) framework to estimate the association of genetically predicted plasma caffeine level and caffeine intake on kidney traits, which comprised glomerular filtration rate estimated from creatinine (eGFRcrea) and glomerular filtration rate estimated from cystatin-c (eGFRcyst), urinary sodium, urinary, blood urea nitrogen (BUN), albumin-creatinine ratio (UACR), and risk of CKD. With this study, we aimed to overcome the limitations of traditional observational methods, including environmental confounding and reverse causation, by employing the MR paradigm. In particular, because the genetic variants used as instrumental variables in MR are randomly allocated at conception, their associations are not typically affected by environmental confounding factors. Additionally, their allocation precedes the exposure and outcome, protecting against reverse causation bias. Further, we consider both caffeine intake and plasma caffeine levels, respectively, as the relevant exposures of interest.

## 2. Materials and Methods

### 2.1. Plasma Caffeine Level Measurement and Data Sources

Genetic association summary statistics for plasma caffeine level were retrieved from a meta-analysis of six genome-wide association studies (GWAS) on caffeine metabolites: the Prospective Study of the Vasculature in Uppsala Seniors (PIVUS), the Study of Health in Pomerania TREND (SHIP-TREND), the Swiss Kidney Project on Genes in Hypertension (SKIPOGH), TwinGene, TwinsUK and the Uppsala Longitudinal Study of Adult Men (ULSAM), including a total of 9876 European ancestry individuals [17]. Plasma caffeine level was measured with ultraperformance liquid chromatography-tandem mass spectrometry (UPLC-MS/MS) in ULSAM, PIVUS, TwinGene, SKIPOGH cohorts, or ultra-performance liquid chromatography-electrospray tandem mass spectrometry (UPLC-ESI-MS/MS) in SHIP-TREND and TwinsUK cohorts. Details are shown in Appendix A.

### 2.2. Definition of Caffeine Intake and Data Sources

Caffeine intake summary statistics were derived from a GWAS meta-analysis including >47,000 individuals of European descent from 5 studies (Atherosclerosis Risk in Communities (ARIC), the Prostate, Lung, Colorectal, and Ovarian Cancer Screening Trial (PLCO), the Nurses’ Health Study (NHS), the Health Professionals Follow-Up Study (HPFS), Women’s Genome Health Study (WGHS)) [16]. In each study, the caffeine intake was assessed with a food frequency questionnaire (FFQ). Raw caffeine intake values were skewed across studies and adjusted for age, sex, case–control status, study site, smoking, and study-specific eigenvectors. Details are shown in Appendix A.

### 2.3. Definition of the Outcomes and Data Sources

Summary statistics related to estimated glomerular filtration rate (eGFR), urinary albumin-creatinine ratio (UACR), blood urea nitrogen (BUN), and chronic kidney disease (CKD) were retrieved from CKDGen consortium GWASs [18,19,20,21]. The eGFR data were obtained from a meta-analysis of CKDGen and UK Biobank [19]. The creatinine-based eGFR (eGFRcrea) was available for >1 million individuals of European ancestry and was computed using the Chronic Kidney Disease Epidemiology Collaboration (CKD-EPI) formula [22]. The serum cystatin-C-based eGFR (eGFRcyst) was calculated using the CKD-EPI formula [22], and the summary statistics were available only in the trans-ancestry population (n = 460,826). As described in the original study, both eGFR estimations were winsorized between 15 and 200 mL/min/1.73 m^2^ and log transformed [19]. Urinary sodium was obtained from 326,831 individuals in UK Biobank GWAS summary statistics [23]. BUN was computed by multiplying blood urea (mg/dL) by 2.8 [20]. UACR (mg/g) was calculated as the ratio of urinary albumin (mg/L) and urinary creatinine (in mg/dL) multiplied by 100 [21]. Both measures were log-transformed. CKD (defined as eGFRcrea below 60 mL/min/1.73 m^2^) was available for 41,395 cases and 439,303 controls [20]. Details are shown in Appendix A.

### 2.4. Statistical Analysis

As instruments, we selected single nucleotide polymorphisms (SNPs) within a 100 kb window of the *CYP1A2* and *AHR* gene regions that were associated with plasma caffeine levels [17] or caffeine intake [16], respectively. These genes were chosen because of their role in increasing the metabolism of caffeine [17] and have been used as instruments for plasma caffeine level and caffeine intake in previous MR analyses [24,25]. The SNPs within each locus were in linkage disequilibrium (r^2^ ranging from 0.21 to 0.96 in European populations). Therefore, we selected the strongest signal at each locus, that is, rs4410790 at AHR and rs242297 at CYP1A2 for plasma caffeine level and rs4410790 at *AHR* and rs2470893 at *CYP1A2* for caffeine intake, with details shown in Table 1.

Two-sample summary data MR analysis was performed using the random effects inverse-variance weighted method. This generates Wald ratios as the variant-outcome association divided by the variant-exposure association, with MR standard errors estimated as the standard error of the variant-outcome association divided by the variant-exposure association. Estimates for the two instrument variants were pooled via random-effects inverse-variance meta-analysis. The statistical analysis and figure were generated using R statistical software version 4.3.1.

## 3. Results

The MR analysis identified that a 1-SD increase in genetically predicted plasma caffeine level was associated with a decrease in eGFR (beta [95% confidence interval] = −0.025 [−0.028,−0.022]; *p* = 4.1 × 10^−53^ for log(eGFRcrea); −0.018 [−0.025,−0.011]; *p* = 1.6 × 10^−6^ for log(eGFRcyst)), and log(UACR) (−0.179 [−0.215,−0.143]; *p* = 1.7 × 10^−22^). Genetically predicted plasma caffeine levels were also associated with higher urinary sodium levels (0.149 [0.115,0.170]; *p* = 1.6 × 10^−24^) and BUN (0.04 [0.025,0.055]; *p* = 1.9 × 10^−7^).

In contrast, higher genetically predicted caffeine intake associated with an increase in both eGFR measures (0.020 [0.014,0.015]; *p* = 1.8 × 10^−11^ for log(eGFRcrea) and 0.013 [0.03,0.02]; *p* = 0.01, for log(eGFRcyst)), and log(UACR) (0.163 [0.120,0.206]; *p* = 1.6 × 10^−3^). A 1-SD increase in genetically predicted caffeine intake was also associated with a lower risk of CKD (odds ratio [95% confidence interval]: 0.84 [0.75,0.94]; *p* = 0.003), lower urinary sodium (−0.125 [−0.158,−0.093]; *p* = 2.6 × 10^−14^), and BUN (−0.034 [−0.064,−0.005]; *p* = 0.023). A graphical summary of the results is shown in Figure 1.

## 4. Discussion

This study supports evidence of a detrimental effect of higher genetically predicted plasma caffeine levels on kidney function according to two different measures of eGFR. Additionally, the results showed adverse associations of genetically predicted plasma caffeine levels with biological markers of CKD progression, including urinary sodium and BUN [26,27,28], but not UACR. Our results provide evidence supporting similar detrimental effects to those found in experimental studies investigating long-term caffeine consumption on rats and mice carrying less genetic variability [4,6]. It was suggested that caffeine modulates changes in eGFR via diuresis and natriuresis through binding adenosine receptors, interferes with the anti-inflammatory effects of adenosine, and stimulates some of the key proliferative mechanisms involved in glomerular remodeling and sclerosis [3,29]. It is interesting to note that we found no evidence of a detrimental effect of genetically predicted plasma caffeine levels on albuminuria, reinforcing the idea that the potentially detrimental effects of caffeine do not act via an increase in glomerular capillary hydraulic pressure which would cause glomerular damage and therefore albuminuria, but via other mechanisms such as those mentioned in the experimental studies cited above.

In contrast, a protective effect of higher genetically predicted caffeine intake on kidney function was found. The discrepancy is likely attributable to faster metabolizers of caffeine requiring a greater caffeine intake to achieve the same stimulant effect. In addition, this could potentially explain the beneficial or absence of association between coffee intake and CKD shown in the literature because all studies assessed coffee consumption rather than plasma caffeine levels as the exposure. Indeed, a recent study that investigated caffeine and kidney function provided evidence that coffee has different effects on kidney function in slow versus fast metabolizers [15]. When assessing the effect of genetically predicted amount of caffeine intake on kidney function, we found, however, a potential protective effect similar to that found in most prospective observational studies and meta-analyses [9,10,11,12,13,14]. This result was also similar to a recent MR study conducting a two-sample MR of genetically predicted coffee consumption using also the CKDGen Consortium GWAS [25]. They selected 25 SNPs, however, including the ones we considered, and found that an extra cup of coffee per day conferred a protective effect against CKD. The discrepancy in the effect of caffeine on kidney function when assessing caffeine levels or coffee consumption as exposure in our study and the literature is of high importance. This provides evidence that there may be an interaction between the amount of coffee consumption and genetic predisposition to metabolize this caffeine on the effect of caffeine on kidney function. Future research exploring the effect of coffee on an outcome should take this into account.

However, with our current study, it is not possible to distinguish whether the observed evidence for the effects of caffeine on kidney traits is driven by the harmful effects of plasma caffeine in slower metabolizers or the protective effects of greater caffeine intake in faster metabolizers. The former may be explained by the diuretic properties of caffeine having harmful effects on kidney function, while the latter may be due to the protective effects of greater caffeine intake through improved hydration status, given that the majority of caffeine intake is via beverages. Further research is warranted to disentangle these findings, particularly given the widespread use of caffeine-containing drinks and the increasing burden of kidney disease.

The MR design used in our study was a strength for reducing bias from environmental factors when assessing associations to infer causal effects. Moreover, because genetic variants are fixed at conception, reverse causation is unlikely. A limitation of MR is pleiotropic effects (i.e., the plasma caffeine level SNPs affect the outcome not only through the exposure), which was minimized in this study by selecting genetic instruments with effects that plausibly act directly on the trait in question (i.e., genes encoding enzymes with an established role in caffeine metabolism, either directly via the CYP1A2 enzyme or indirectly (AHR) by the regulation of CYP1A2 expression. A limitation of this study is that our findings might not be generalizable to a population of non-European ancestry because all genetic summary statistics are from European ancestry. Nevertheless, a previous study confirmed that *AHR* and *CYP1A2* polymorphisms are associated with caffeine consumption in a non-European population, which provides an argument that the genetic predictors of plasma caffeine level we used might be similar in non-European ethnicity individuals [30]. Another potential shortcoming of our study is that we could not conduct statistical sensitivity analyses using commonly employed MR methods for detecting possible pleiotropy, such as MR-Egger regression and MR-PRESSO, because those approaches require at least three or more instrumental variables.

## 5. Conclusions

This MR analysis provides evidence that there is an effect of coffee consumption and genetic predisposition to caffeine metabolism on kidney function. It suggests a detrimental effect of genetically predicted higher plasma caffeine levels and a protective effect of a genetically predicted higher amount of caffeine intake on kidney function. Further research is needed to distinguish whether the observed effects on kidney traits are driven by the harmful effects of plasma caffeine in slower metabolizers or the protective effects of greater caffeine intake in faster metabolizers.

## Figures and Tables

**Figure 1 nutrients-15-04422-f001:**
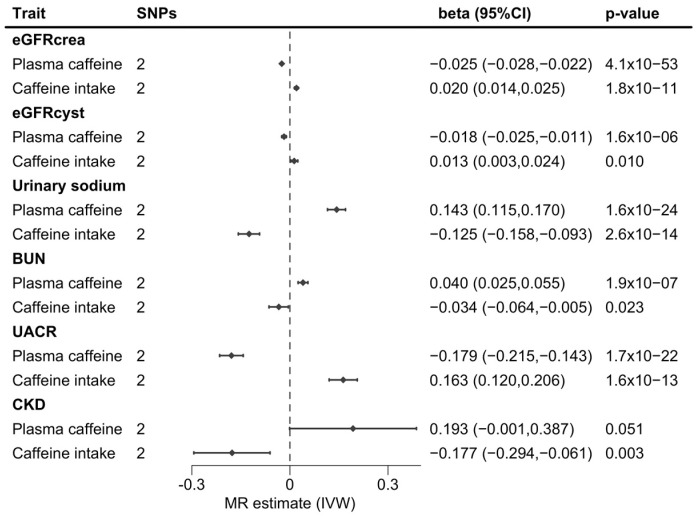
Forest plot of two-sample MR estimates the effects of genetically predicted plasma caffeine levels or caffeine intake using the AHR and CYP1A2 genes on kidney traits. Abbreviation: MR, Mendelian randomization; SNP, single-nucleotide polymorphism; CI, confident interval; eGFRcrea, glomerular filtration rate estimated from creatinine; eGFRcys, glomerular filtration rate estimated from cystatin-c; BUN, blood urea nitrogen; UACR, albumin-creatinine; CKD, chronic kidney disease; IVW, inverse-variance weighted two-sample Mendelian randomization analysis.

**Table 1 nutrients-15-04422-t001:** Association of the genetic variants with the exposures.

Exposure	Effect Allele	Other Allele	Beta	SE	*p*-Value	Gene	EAF
Plasma caffeine							
rs4410790	T	C	0.109	0.015	1.80 × 10^−10^	*AHR*	0.36
rs2472297	C	T	0.150	0.016	1.00 × 10^−17^	*CYP1A2*	0.73
Caffeine intake							
rs4410790	C	T	0.150	0.017	2.36 × 10^−19^	*AHR*	0.38
rs2470893	T	C	0.120	0.016	5.15 × 10^−14^	*CYP1A2*	0.31

Abbreviation: SE: standard error; EAF: effect allele frequency.

## Data Availability

All data used in this study are publicly available from the cited sources.

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
