# Peer review of "Caffeine Intake, Plasma Caffeine Level, and Kidney Function: A Mendelian Randomization Study"

_nutrients, 2023, doi:10.3390/nu15204422_

Round 1

Reviewer 1 Report

On line 73 the authors mention incidentally the “reverse causation” concept without being explicitly defined earlier. It would be useful to define it explicitly in the introduction above its mentioning here.

 Section 2.4: the authors do not provide any information on the statistical approach and software used, they only justify the two SNPs used in the MR. Minimal information on the statistical information and software should be mentioned here (e.g. how were betas computed, how were the 95% IC estimated), how were the p values estimated/computed, what covariates were used (if any) etc, what software was used to generate the forest plot in Figure 1…

Author Response

On line 73 the authors mention incidentally the “reverse causation” concept without being explicitly defined earlier. It would be useful to define it explicitly in the introduction above its mentioning here.

This has now been explained by instead writing:

“With this study we aimed to overcome the limitation of traditional observational methods, including environmental confounding and reverse causation, by employing the MR paradigm. Specifically, because the genetic variants used as instrumental variables in MR are randomly allocated at conception, their associations are not typically affected by environmental confounding factors and additionally their allocation precedes the exposure and outcome, protecting against reverse causation bias.”

Section 2.4: the authors do not provide any information on the statistical approach and software used, they only justify the two SNPs used in the MR. Minimal information on the statistical information and software should be mentioned here (e.g. how were betas computed, how were the 95% IC estimated), how were the p values estimated/computed, what covariates were used (if any) etc, what software was used to generate the forest plot in Figure 1…

Details of covariate adjustments are provided in section 2.2. We now include the requested information in section 2.4:

“Two-sample summary data MR analysis was performed using the random effects inverse-variance weighted method. This generates Wald ratios as the variant-outcome association divided by the variant-exposure association, with MR standard errors estimated as the standard error of the variant-outcome association divided by the variant-exposure association. Estimates for the two instrument variants were pooled by random-effects inverse-variance meta-analysis. The statistical analysis was performed using R statistical software and the MendelianRandomization package.”

Reviewer 2 Report

Manuscript ID: nutrients-2651420  

Manuscript Title: Caffeine intake, plasma caffeine levels and kidney function: a

Mendelian randomization study

Journal: Nutrients  

 The manuscript is meaningful. However, it needs for major revision before it is considered for publication. My comments are as follows.

Title Section

1. L 2: Revise “plasma caffeine levels” to “plasma caffeine level”.

 Abstract Section

2. L 24-25: Regarding “Caffeine is a psychoactive substance widely consumed worldwide mainly through sources such as coffee, tea, and soft drinks”, it is not appropriate to list “soft drinks” as a kind of source of caffeine. “Soft drinks” are artefacts. 

3. L 25: Revise “Effects on kidney” to “The effects of caffeine on kidney”.

4. L 26: Revise “the CYP1A2 and AHR genes” to “CYP1A2 and AHR genes”.

5. L 30: Revise “>47,000” to “> 47,000”.

6. L 37: Revise “caffeine containing drinks” to “drinks containing caffeine”. Is the change right?

7. L 40: Revise “Mendelian randomization (MR)” to “Mendelian randomization”.

8. L 40: Revise “eGFR” to “estimated glomerular filtration rate”.

Introduction Section

9. L 43: Revise “Caffeine is a central nervous system stimulant present” to “Caffeine, a central nervous system stimulant, presents”.

10. L 44: Revise “beans, that is” to “beans, and is”.

11. L 44: Revise “added to” to “added in”.

12. L 46: Revise “adverse effects” to “adverse effects of caffeine”.

13. L 53-55: Regarding “Recent prospective observational studies and meta-analyses observed a lower risk of developing chronic kidney disease (CKD) with increasing coffee consumption [9–14]., references “[9–14]” published between 2010 and 2021, and does not support “Recent prospective observational studies.

14. L 59-61: Regarding “the risks of albuminuria and hyperfiltration were over 2-fold higher among slow metabolizers of caffeine who consumed more than 3 cups of coffee per day, whereas there was no difference among fast metabolizers [15].”. Please clarify which populations the authors compared with. 

15. Please note the consistency of key terms. such as “glomerula filtration rate estimated from creatinine (eGFRcrea)” on line 68 and estimated glomerular filtration rate (eGFR) on line 31.

16. L 68-69: “cystatin-c (eGFRcyst)”?

17. L 70-72: Regarding “Genetic variants are assigned randomly during meiosis independently of environmental confounders and are fixed at conception, thus not affected by outcomes.”. Under what circumstances are genetic variants influenced by outcomes?

Materials and Methods Section

18. L 77: Revise “Definition of plasma caffeine” to “The definition of plasma caffeine”.

19. L 77: Regarding 2.1. Definition of plasma caffeine level and data sources section, The authors did not provide any relevant information on “Definition of plasma caffeine level”. 

20. L 87: Revise “Details are” to “Details were”.

21. L 104: “was computed calculated”?

22.  L 107: “winsorized”?

23.  L 117: Revise “[16] respectively” to “[16], respectively”.

24. L 118: Revise “, [17] and” to “[17], and”.

Results Section

25. L 140: Revise “is shown” to “was shown”.

 Discussion Section

26. L 152-154: Regarding “This result was consistent with the detrimental causal estimates of genetically predicted plasma level on others biologial markers related to CKD progression (urinary sodium and BUN) [26–28],”, Please provide specific information on “This result.

27. L 157: Revise “caffeine modulates” to “caffeine modulated”.

28. L 171-172: Revise “study that investigated” to “study investigated”.

29. L 196: Revise “design of our study” to “design used in our study”.

30. L 208: Revise “plasma caffeine concentration” to “plasma caffeine level”.

 Conclusion Section

31. L 214: Revise “This MR analysis provides” to “This MR analysis provided”.

32. L 214: Revise “there is an effect” to “there was an effect”.

33. L 215: Revise “It suggests” to “It suggested”.

34. L 216: Revise “plasma caffeine levels” to “plasma caffeine level”.

There are some minor grammar issues that need to be corrected. Please see my comments.

Author Response

Title Section

  1. L 2: Revise “plasma caffeine levels” to “plasma caffeine level”.

Change made.

Abstract Section

  1. L 24-25: Regarding “Caffeine is a psychoactive substance widely consumed worldwide mainly through sources such as coffee, tea, and soft drinks”, it is not appropriate to list “soft drinks” as a kind of source of caffeine. “Soft drinks” are artefacts.

Change made.

  1. L 25: Revise “Effects on kidney” to “The effects of caffeine on kidney”.

Change made.

  1. L 26: Revise “the CYP1A2 and AHR genes” to “CYP1A2 and AHR genes”.

Change made.

  1. L 30: Revise “>47,000” to “> 47,000”.

Change made.

  1. L 37: Revise “caffeine containing drinks” to “drinks containing caffeine”. Is the change right?

Change made.

  1. L 40: Revise “Mendelian randomization (MR)” to “Mendelian randomization”.

Change made.

  1. L 40: Revise “eGFR” to “estimated glomerular filtration rate”.

Change made.

Introduction Section

  1. L 43: Revise “Caffeine is a central nervous system stimulant present” to “Caffeine, a central nervous system stimulant, presents”.

Change made.

  1. L 44: Revise “beans, that is” to “beans, and is”.

Change made.

  1. L 44: Revise “added to” to “added in”.

Change made.

  1. L 46: Revise “adverse effects” to “adverse effects of caffeine”.

Change made.

  1. L 53-55: Regarding “Recent prospective observational studies and meta-analyses observed a lower risk of developing chronic kidney disease (CKD) with increasing coffee consumption [9–14].”, references “[9–14]” published between 2010 and 2021, and does not support “Recent prospective observational studies”.

Change made.

  1. L 59-61: Regarding “the risks of albuminuria and hyperfiltration were over 2-fold higher among slow metabolizers of caffeine who consumed more than 3 cups of coffee per day, whereas there was no difference among fast metabolizers [15].”. Please clarify which populations the authors compared with.

Change made.

  1. Please note the consistency of key terms. such as “glomerula filtration rate estimated from creatinine (eGFRcrea)” on line 68 and “estimated glomerular filtration rate (eGFR)” on line 31.

Change made.

  1. L 68-69: “cystatin-c (eGFRcyst)”?

Change made.

  1. L 70-72: Regarding “Genetic variants are assigned randomly during meiosis independently of environmental confounders and are fixed at conception, thus not affected by outcomes.”. Under what circumstances are genetic variants influenced by outcomes?

We have clarified this text to reduce ambiguity. A detailed discussion of mechanisms by which the environment can affect germline variation is beyond the scope of this article.

Materials and Methods Section

  1. L 77: Revise “Definition of plasma caffeine” to “The definition of plasma caffeine”.

Change made.

  1. L 77: Regarding “2.1. Definition of plasma caffeine level and data sources” section, The authors did not provide any relevant information on “Definition of plasma caffeine level”.

Change made.

  1. L 87: Revise “Details are” to “Details were”.

This change is not correct, we kept the original version.

  1. L 104: “was computed calculated”?

Change made.

  1. L 107: “winsorized”?

This is a statistical term.

  1. L 117: Revise “[16] respectively” to “[16], respectively”.

Change made.

  1. L 118: Revise “, [17] and” to “[17], and”.

Change made.

Results Section

  1. L 140: Revise “is shown” to “was shown”.

This change is not correct, the change was not made.

Discussion Section

  1. L 152-154: Regarding “This result was consistent with the detrimental causal estimates of genetically predicted plasma level on others biologial markers related to CKD progression (urinary sodium and BUN) [26–28],”, Please provide specific information on “This result”.

Change made.

  1. L 157: Revise “caffeine modulates” to “caffeine modulated”.

This change is not correct, no change made.

  1. L 171-172: Revise “study that investigated” to “study investigated”.

This change is not correct, no change made.

  1. L 196: Revise “design of our study” to “design used in our study”.

This change is not correct, no change made.

  1. L 208: Revise “plasma caffeine concentration” to “plasma caffeine level”.

Change made.

Conclusion Section

  1. L 214: Revise “This MR analysis provides” to “This MR analysis provided”.

This change is not correct, no change made.

  1. L 214: Revise “there is an effect” to “there was an effect”.

This change is not correct, no change made.

  1. L 215: Revise “It suggests” to “It suggested”.

This change is not correct, no change made.

  1. L 216: Revise “plasma caffeine levels” to “plasma caffeine level”.

Change made.

Round 2

Reviewer 2 Report

Manuscript ID:  nutrients-2651420-V2 

Manuscript Title: Caffeine intake, plasma caffeine levels and kidney function: a

Mendelian randomization study

Journal: Nutrients  

The quality of the mst has been improved significantly, but it is recommended that the authors solve the following minor issues before the ms is accepted.

1. L 35-36: Revise “the observed effects on kidney” to “the found effects on kidneys”.

2. L 45: Revise “the phosphodiesterase enzyme” to “phosphodiesterase enzyme”.

3. L 63: Revise “the gene expression” to “gene expression”.

4. L 64: Revise “Heritability of coffee” to “The heritability of coffee”.

5. L 71: Revise “(UACR) and” to “(UACR), and”.

6. L 84: Revise “plasma caffeine levels” to “plasma caffeine level”.

7. L 93, 119: Revise “Details are” to “Details were”.

8. L 112: Revise “n=460,826” to “n = 460,826”.

9. L 113: Revise “ml/min/1.73m2” to “mL/min/1.73 m2”. Is the change right?

10. L 116: Revise “mg/l” to “mg/L”.

11. L 117: Revise “mg/dl” to “mg/dL”.

12. L 118: Please rewrite “60 ml/min-1 per 1.73m2”.

13. L 125: Revise “[24,25]” to “[24, 25]”.

14. L 153: Revise “is shown” to “was shown”.

15.  L 168: Revise “Our results provide” to “Our results provided”.

16. L 171: Revise “caffeine modulates” to “caffeine modulated”.

17. L 172: Revise “adenosine and stimulates” to “adenosine, and stimulated”.

18. L 187-188: Revise “coffee has” to “coffee had”.

19. L 197: Revise “This provides evidence” to “This provided evidence”.

20. L 221: Revise “which provides” to “which provided”.

21. L 229: Revise “This MR analysis provides” to “This MR analysis provided”.

22. L 229: Revise “there is” to “there was”.

23. L 230: Revise “It suggests” to “It suggested”.

24. L 231: Revise “plasma caffeine levels” to “plasma caffeine level”.

Minor editing of English language required

Author Response

Manuscript ID:  nutrients-2651420-V2

Manuscript Title: Caffeine intake, plasma caffeine levels and kidney function: a

Mendelian randomization study

Journal: Nutrients  

The quality of the mst has been improved significantly, but it is recommended that the authors solve the following minor issues before the ms is accepted.

We thank the Reviewer for their considered and helpful feedback.

  1. L 35-36: Revise “the observed effects on kidney” to “the found effects on kidneys”.

This has been revised to “the observed effects on kidney function”.

  1. L 45: Revise “the phosphodiesterase enzyme” to “phosphodiesterase enzyme”.

Change made.

  1. L 63: Revise “the gene expression” to “gene expression”.

Change made.

  1. L 64: Revise “Heritability of coffee” to “The heritability of coffee”.

Change made.

  1. L 71: Revise “(UACR) and” to “(UACR), and”.

Change made.

  1. L 84: Revise “plasma caffeine levels” to “plasma caffeine level”.

Change made.

  1. L 93, 119: Revise “Details are” to “Details were”.

“Details are” is correct and no change was made. These details are presently shown, so past tense is not required.

  1. L 112: Revise “n=460,826” to “n = 460,826”.

Change made.

  1. L 113: Revise “ml/min/1.73m2” to “mL/min/1.73 m2”. Is the change right?

Change made to mL/min/1.73m2.

  1. L 116: Revise “mg/l” to “mg/L”.

Change made.

  1. L 117: Revise “mg/dl” to “mg/dL”.

Change made.

  1. L 118: Please rewrite “60 ml/min-1 per 1.73m2”.

Change made.

  1. L 125: Revise “[24,25]” to “[24, 25]”.

Change made.

  1. L 153: Revise “is shown” to “was shown”.

“is shown” is correct and no change was made. This detail is presently shown, so past tense is not required.

  1. L 168: Revise “Our results provide” to “Our results provided”.

It is correct as currently written because our results currently show this. No change made.

  1. L 171: Revise “caffeine modulates” to “caffeine modulated”.

It is correct as already written because this statement is currently true. No change made.

  1. L 172: Revise “adenosine and stimulates” to “adenosine, and stimulated”.

It is correct as already written because this statement is currently true. No change made.

  1. L 187-188: Revise “coffee has” to “coffee had”.

It is correct as already written because this statement is currently true. No change made.

  1. L 197: Revise “This provides evidence” to “This provided evidence”.

It is correct as already written because this statement is currently true. No change made.

  1. L 221: Revise “which provides” to “which provided”.

It is correct as already written because this statement is currently true. No change made.

  1. L 229: Revise “This MR analysis provides” to “This MR analysis provided”.

It is correct as already written because this statement is currently true. No change made.

  1. L 229: Revise “there is” to “there was”.

It is correct as already written because this statement is currently true. No change made.

  1. L 230: Revise “It suggests” to “It suggested”.

It is correct as already written because this statement is currently true. No change made.

  1. L 231: Revise “plasma caffeine levels” to “plasma caffeine level”.

Change made.